# Analysis of Heavy Metal Content in Conventional and Herbal Toothpastes Available at Maltese Pharmacies

**Andrew Vella [1] and Everaldo Attard [2],\***

[1] Department of Pharmacy, Faculty of Medicine and Surgery, University of Malta, Msida MSD 2080, Malta; andrewvella94@gmail.com

[2] Division of Rural Sciences and Food Systems, Institute of Earth Systems, University of Malta, Msida MSD 2080, Malta

\* Correspondence: everaldo.attard@um.edu.mt

**Abstract:** Although toothpastes are considered as topical cosmetics that are not normally ingested, it is evident that they may contribute to the introduction of heavy metals and xenobiotics through buccal and gastrointestinal absorption. The purpose of this study was to determine the potential presence of metals and polyphenols in conventional, children's and herbal toothpastes. Metal analysis was conducted by using the Microwave Plasma-Atomic Emission Spectrometer and the total polyphenolic content was determined by using the Folin–Ciocalteu test. Results showed that cadmium and mercury were absent in all toothpastes while zinc and tin exhibited high values. This was because the latter two metals are incorporated as part of the ingredients. In the case of polyphenols, the highest value was obtained in one of the samples from the children's toothpaste category while the lowest value was obtained from the conventional toothpaste category. Lead and nickel were two other metals that fell outside the limits for EU and US standards. Most of these limits are usually applicable to topical cosmetic products or food products. However, these may not adequately cover oral hygiene products, such as toothpastes.

**Keywords:** heavy metals; total polyphenolic content; international standards; toothpastes

## 1. Introduction

The presence of heavy metals and xenobiotics are not normally considered as a primary concern in cosmetics. Although in most cases, cosmetics are applied by a category of a population, the use of toothpastes on a daily basis to maintain oral hygiene is cosmopolitan. Consequently, the daily constant ingestion of heavy metals and xenobiotics raises safety issues [1]. This is because many countries do not regulate the presence of these heavy metals and xenobiotics in cosmetic products. Such substances may be a result of unintentional addition to toothpastes, primarily as impurities with other ingredients or as a result of manufacturing [2]. Whereas heavy metals such as lead (Pb), cadmium (Cd), nickel (Ni) and arsenic (As) may be considered as detrimental to human health, other metals such as zinc (Zn), copper (Cu) and iron (Fe) may be regarded as necessary in small quantities and within certain limits. Although, the cosmetic industry aims to manufacture products of high quality through good manufacturing practices (GMPs), it is evident that these metals and xenobiotics may be present in high quality products. Regulatory bodies, worldwide, sought to address such issues by establishing guidelines and rules to safeguard the consumer [3]. On the other hand, the presence of xenobiotics may prove to be beneficial [4]. The presence of polyphenols in cosmetic products is not uncommon [5] and their primary function is to act as antioxidants both during the shelf-life of the product but also when the product is applied to the human body.

The aim of this study was to determine the presence of heavy metals in toothpastes available at Maltese pharmacies. This study also investigated whether the presence of these heavy metals falls within the permissible limits of heavy metals for cosmetic products and whether the presence of polyphenols in the toothpaste samples investigated correlated with the amount of heavy metals.

## 2. Materials and Methods

### 2.1. Samples

Nine toothpastes were randomly selected from three different categories; three per category. These included conventional, herbal and children's toothpastes. Although the samples were taken from a pharmacy setting, these products are available at supermarkets, beauty shops and other outlets in Malta and other European countries.

### 2.2. Reagents and Solutions

All reagents used for both metal and polyphenolic content analysis were of analytical grade. For metal analysis, $HNO_3$ (65%) and $H_2O_2$ (34.5%) (Sigma-Aldrich, Germany) were used for the preparation of solutions required for the digestion of samples. Ultrapure water was obtained from a New Human UP900 Integrate (Human Corporation, Republic of Korea), which was used for the preparation of the 5% $HNO_3$ solution and for the dilution of samples after ashing. For metal analysis, a certified standard stock solution (Multielement standard solution 6 for ICP, TraceCERT, Sigma-Aldrich, Taufkirchen, Germany) was used to calibrate the instrument before running the samples. In the analytical procedure, the limit of detection (LOD) was considered as the lowest amount of analyte which can be quantified with a known degree of reliability, while the limit of quantitation (LOQ) refers to the concentration at which quantitative results can be produced at sufficient amount of confidence. These were determined using a statistical approach which is based on measuring replicate blank samples or through the measurement of progressively dilute concentrations of the analyte [6]. The LOD was calculated as 3.3× (SD/m) and the LOQ as 10× (SD/m), where SD is the standard deviation of the blank and m is the slope of the calibration curve. The respective wavelength, linear correlation coefficients ($R^2$), LOD and LOQ of Ag, Cd, Cr, Cu, Hg, Ni, Pb, Sn, Zn, Fe and Mn are illustrated in Table 1.

For the polyphenolic content, the Folin–Ciocalteu reagent, anhydrous $Na_2CO_3$ and gallic acid (Sigma-Aldrich, Taufkirchen, Germany) were used. The Folin–Ciocalteu reagent was prepared as a 10% solution in deionized water and $Na_2CO_3$ as a 1 M solution, while gallic acid was prepared as a set of dilutions, 0, 60, 120, 240, 480 and 960 µg/mL in deionized water.

**Table 1.** The respective SD, linear correlation coefficients ($R^2$), limit of detection (LOD) and limit of quantitation (LOQ) of Ag, Cd, Cr, Cu, Fe, Hg, Mn, Ni, Pb, Sn and Zn.

| Element | Wavelength (nm) | $R^2$ | LOD (mg/kg) | LOQ (mg/kg) |
|---------|-----------------|-------|-------------|-------------|
| Ag | 328.068 | 0.9746 | 0.0549 | 0.1665 |
| Cd | 228.802 | 0.9993 | 0.0067 | 0.0204 |
| Cr | 425.433 | 0.9975 | 0.0005 | 0.0014 |
| Cu | 324.754 | 0.9996 | 0.0007 | 0.0022 |
| Fe | 259.940 | 1.0000 | 0.0037 | 0.0113 |
| Hg | 253.652 | 0.9968 | 0.0789 | 0.2391 |
| Mn | 403.076 | 0.9288 | 0.0042 | 0.0127 |
| Ni | 352.454 | 0.9987 | 0.0056 | 0.0169 |
| Pb | 405.781 | 1.0000 | 0.0169 | 0.0511 |
| Sn | 317.505 | 0.9852 | 0.0375 | 0.1137 |
| Zn | 213.857 | 1.0000 | 0.0301 | 0.0912 |

## 2.3. Preliminary Treatment of Samples

For metal analysis, approximately 1 g of each sample, in triplicates, were weighed for the nine toothpaste samples and placed in porcelain crucibles. Samples were digested using 5 mL of 5% $HNO_3$ and 2 mL of $H_2O_2$, which were added sequentially and left to evaporate on a hot plate at 80–90 °C upon each addition. Following digestion, the samples were ashed in a muffle furnace (Wisetherm, Wisd, Laboratory Instruments, Germany) for 6 h at 500 °C. After ashing, the crucible contents were dissolved in 5 mL of 5% nitric acid. The residues left were filtered and finally the filtrates were volumetrically topped up to 50 mL with de-ionized water.

For the total polyphenolic content, approximately 1g of each toothpaste sample, in triplicates, was weighed and mixed with 20 mL of methanol. The mixture was then ultrasonicated (VWR IP23, Radnor, Pennsylvania, USA) for 10 min prior to analysis.

## 2.4. Measurements

The metals were analyzed in a Microwave Plasma-Atomic Emission Spectrometer (MP-AES 4100, Agilent Technologies Inc., Santa Clara, California, USA). The system was controlled by the Agilent software, MP Expert (version 1.5.1.6821, Agilent Technologies Inc., Santa Clara, California, USA). The instrument settings are illustrated in Table 2.

The total polyphenolic content was analyzed using the Folin–Ciocalteu test [7]. Briefly, triplicates of 10 µL of methanolic extracts were pipetted in wells of a microtiter plate (Nunc, Roskilde, Denmark). To each well, 100 µL of Folin–Ciocalteu reagent and 80 µL of 1M $Na_2CO_3$ were added. Additionally, 10 µL of the gallic acid standards were also pipetted in triplicates together with the reagents. The plate was incubated in the dark at room temperature for 20 min and then analyzed using a microtiter plate reader (SpectroStar-Nano, BMG, Labtech, Ortenberg, Germany) at 750 nm. A calibration curve ($y = mx + c$) with concentration against absorbance was plotted using the gallic acid standards.

**Table 2.** The settings and conditions for the Microwave Plasma-Atomic Emission Spectrometer (MP-AES) (Agilent 4100).

| | |
|---|---|
| Nebulizer | One-neb (concentric) |
| Spray chamber | Cyclonic spray chamber, single pass |
| Calibration Correlation Coefficient Limit | 0.9999 |
| Pump speed (rpm) | 15 |
| Number of replicates | 3 |
| Stabilization time (s) | 15 |
| Uptake time (s) | 15 |

## 2.5. Statistical Methods

The data obtained were analyzed with Prism 5 (GraphPad Software Inc., San Diego, California, USA) and XLSTAT (Microsoft, version 19.4.46756, SAS Institute Inc., Marlow, Buckinghamshire, UK) software. With GraphPad-Prism, ANOVA with the Bonferroni post-hoc test was performed on all triplicate values for the heavy metals and total polyphenols analyzed in this study. XLSTAT was used to conduct the Pearson correlation on sample data sets. This was conducted to determine any correlation or divergences between parameters of the toothpaste samples.

## 3. Results and Discussion

### 3.1. Heavy Metal Analysis

Although the term 'heavy metals' poorly describes the nature of metals in scientific terms [8], in this present study, both toxic metals (Ag, Cd, Hg, Pb and Sn) and essential microminerals (Cr, Cu, Ni, Zn, Fe and Mn) were grouped under this general definition. Previous studies showed that metals such as Pb and Cu are present in toothpastes and other cosmetic products [1,9]. These impurities may reach

the general circulation in the body leading to detrimental effects [10,11]. Regulatory bodies provide guidance on a restricted range of heavy metals. Table 3 provides an insight into the regulatory bodies and the corresponding limits for metals. Consequently, this study focused on a wider range of metals, the results of which are shown in Table 4.

**Table 3.** International standards showing the maximum heavy metal content (ppm) in cosmetics and toothpastes.

| Available International Standards | Lead | Cadmium | Mercury | Nickel | Chromium |
|---|---|---|---|---|---|
| EU* [12] | 0.1 | 0.05 | - | - | - |
| WHO [13] | 2 | 2 | - | - | - |
| Canada [14] | 10 | 3 | 3 | - | - |
| Germany [15] | 0.5 | 0.1 | 0.1 | - | - |
| US FDA [16] | 10 | - | 1 | - | 50 |

*in foodstuffs (oral intake).

### 3.1.1. Silver

Silver is a naturally occurring element with particular properties. Apart from industrial applications, Ag is used in medicine for the treatment of burns as a topical antibacterial agent in combination with sulfadiazine. Exposure to Ag occurs through various routes into the body; primarily by inhalation and by skin contact. One of the major health effects linked with exposure to Ag is the development of irreversible pigmentation of the eyes and/or the skin. Overexposure to Ag can lead to accumulation in the skin, kidney, liver and other tissue in the body. The highest concentration of Ag obtained in this study was within the conventional category of toothpastes, with a value of 5.29 ppm, while the lowest mean concentration was obtained with the children's toothpastes (2.00 ppm) ($p < 0.001$). Although, to date, Ag has not been investigated in cosmetic products, including toothpastes, it is well known that Ag is present in dental products in the form of silver diamine fluoride, which potentially arrests dental caries [17].

### 3.1.2. Chromium

As the seventh most abundant element, Cr occurs naturally by the burning of oil, pigment oxidants, oil well drilling and other sources. In this study, the highest concentration of Cr was 7.35 ppm from the conventional group ($p < 0.001$, compared to the rest), while the lowest concentration was obtained from the children's category with a concentration of 0.28 ppm. The majority of values obtained in this study for Cr levels fell within the 1.0 ppm maximum concentration suggested by the WHO [18]. Chromium is significantly used in topical cosmetic products and consequently it is one of the most quoted heavy metals [18–21]. This was also evident through a survey on cosmetics assessed by the US FDA [22]. Chromium and its salts are amongst the prohibited substances in cosmetics within the EU Cosmetics Regulation (Reg. EC. 1223/2009) [23]. Due to its mutagenic properties, chromium (VI) is categorized as a group 1 human carcinogen by the International Agency for the Research on Cancer [24].

**Table 4.** Mean value of heavy metal content for the nine toothpaste samples. All values are expressed as the mean ± SEM.

| Toothpaste Category and Sample No. | Mean Value Ag (ppm) | Mean Value Cd (ppm) | Mean Value Cr (ppm) | Mean Value Cu (ppm) | Mean Value Hg (ppm) | Mean Value Ni (ppm) | Mean Value Pb (ppm) | Mean Value Sn (ppm) | Mean Value Zn (ppm) | Mean Value Fe (ppm) | Mean Value Mn (ppm) |
|---|---|---|---|---|---|---|---|---|---|---|---|
| Conventional 1 | 5.29 ± 0.018 | 0.00 ± 0.000 | 1.42 ± 0.012 | 1.35 ± 0.006 | 0.00 ± 0.000 | 1.96 ± 0.007 | 3.26 ± 0.152 | 134.6 ± 0.253 | 1842 ± 3.550 | 1.76 ± 0.012 | 0.20 ± 0.015 |
| Conventional 2 | 3.36 ± 0.017 | 0.00 ± 0.000 | 7.35 ± 0.019 | 3.68 ± 0.003 | 0.00 ± 0.000 | 1.31 ± 0.006 | 8.83 ± 0.019 | 9671 ± 8.400 | 2417 ± 0.859 | 17.68 ± 0.0088 | 0.72 ± 0.003 |
| Conventional 3 | 3.26 ± 0.035 | 0.00 ± 0.000 | 1.35 ± 0.026 | 0.84 ± 0.003 | 0.00 ± 0.000 | 1.34 ± 0.020 | 2.37 ± 0.064 | 104.7 ± 1.47 | 7.80 ± 1.69 | 12.33 ± 0.000 | 0.66 ± 0.023 |
| Herbal 4 | 5.12 ± 0.020 | 0.00 ± 0.000 | 0.72 ± 0.010 | 1.40 ± 0.003 | 0.00 ± 0.000 | 0.43 ± 0.0033 | 12.04 ± 0.079 | 178.3 ± 0.412 | 2.90 ± 0.072 | 4.50 ± 0.058 | 0.52 ± 0.006 |
| Herbal 5 | 2.23 ± 0.089 | 0.00 ± 0.000 | 0.67 ± 0.012 | 1.26 ± 0.006 | 0.00 ± 0.000 | 1.55 ± 0.021 | 2.23 ± 0.050 | 86.69 ± 1.280 | 3.66 ± 0.051 | 7.84 ± 0.017 | 1.70 ± 0.015 |
| Herbal 6 | 3.10 ± 0.081 | 0.00 ± 0.000 | 0.65 ± 0.015 | 1.16 ± 0.000 | 0.00 ± 0.000 | 1.99 ± 0.032 | 4.33 ± 0.133 | 110.0 ± 0.386 | 6.32 ± 0.076 | 9.68 ± 0.009 | 1.54 ± 0.020 |
| Children 7 | 2.00 ± 0.015 | 0.00 ± 0.000 | 0.4 ± 0.023 | 0.81 ± 0.000 | 0.00 ± 0.000 | 1.73 ± 0.013 | 2.64 ± 0.117 | 82.99 ± 0.307 | 0.31 ± 0.01 | 8.52 ± 0.017 | 2.07 ± 0.031 |
| Children 8 | 3.44 ± 0.029 | 0.00 ± 0.000 | 0.28 ± 0.010 | 0.73 ± 0.003 | 0.00 ± 0.000 | 1.15 ± 0.009 | 4.76 ± 0.062 | 104.4 ± 0.113 | 3.12 ± 0.074 | 10.20 ± 0.009 | 1.13 ± 0.027 |
| Children 9 | 3.82 ± 0.052 | 0.00 ± 0.000 | 0.86 ± 0.009 | 1.37 ± 0.003 | 0.00 ± 0.000 | 2.54 ± 0.035 | 4.72 ± 0.065 | 99.82 ± 0.798 | 0.00 ± 0.000 | 7.56 ± 0.007 | 1.20 ± 0.023 |

### 3.1.3. Copper

Copper is found in all plants and animals and is known as an essential nutrient in small amounts both for humans and animals. There are various sources of Cu contamination in the environment such as mining, copper refining and industries. Cu is beneficial to humans as it is a component of certain enzymes required for metabolic functions. Additionally, low amounts of Cu intake may lead to deficiencies and adverse health effects such as anemia, defects in connective tissue, and reduced white blood cell production, amongst others. Acute short-term poisoning with Cu can lead to temporary gastrointestinal distress. Higher levels of exposure can lead to a destruction of red blood cells, which can even result in anemia. Mammals possess mechanisms which regulate the copper stores in the body making them protected from elevated levels of dietary copper. However, overexposure to this heavy metal can have negative effects to the kidneys and liver. In this study, the highest concentration of Cu was 3.68 ppm from the conventional group, while the lowest concentration was obtained from the children's category with a concentration of 0.73 ppm ($p < 0.001$). The lowest concentration in this study exceeded the maximum Cu level of 0.603 ppm in other cosmetic products [20]. The United States Environmental Protection Agency (EPA) categorizes Cu within group D as there is inadequate evidence to classify it as a potential carcinogen [25].

### 3.1.4. Nickel

Low levels of Ni occur naturally in the environment. Humans are exposed to Ni through various sources such as contaminated water and food, detergents, and also skin contact with Ni- contaminated materials. Nickel is essential for humans in relatively small amounts. However, like most metals, elevated levels of exposure pose adverse effects on human health, which include higher chances of development of various cancers, birth defects, skin rashes and asthma. The highest Ni concentrations were observed in children's toothpastes (2.54 ppm), while the lowest values were observed in the herbal toothpastes (0.43 ppm). There was a significant difference between the two groups ($p < 0.001$). The presence of Ni was found in various cosmetic products with levels ranging from 0.093 and 24.03 ppm [18,20,21]. In a survey on cosmetics assessed by the USFDA, Ni was one of the metals most commonly used in cosmetic products [22]. Nickel-containing alloys are widely used in dental car, particularly in dental restorations.

### 3.1.5. Lead

Lead is known to be one of the most common natural substances on earth. The use of Pb in industry is associated with various processes. Lead does not have any known functions within the human body. Lead toxicity is typically associated with the working population. Although governments have executed mitigation strategies to reduce Pb exposure, childhood Pb poisoning is still considered as a public health concern on a worldwide basis. The clinical manifestations associated with lead toxicity vary from subclinical features such as headache, abdominal pain, loss of appetite to more serious life-threatening complications including renal dysfunction, paralysis and brain damage. The adverse effects associated with Pb are mainly linked with its presence in bone [26]. The analysis for Pb showed that the highest and lowest values were obtained from the herbal toothpaste category (2.23 and 12.04 ppm) ($p < 0.001$). Several authorities provide limitations for the presence of Pb in cosmetics and foodstuffs [12–16]. Most values for Pb obtained in this study exceeded the limits for Pb proposed by various authorities. However, the range of Pb values obtained in this study falls within the range reported by various research groups in cosmetic products (0.205–23.575 ppm) [18,20,21,27]. Apparently, Pb is not intentionally added in cosmetic products but it may be included with the mineral component of the product [22].

### 3.1.6. Tin

Due to the fact that inorganic Sn compounds quickly enter and leave the body after inhalation or ingestion, Sn does not usually lead to harmful effects. However, research studies indicated that humans who swallowed large quantities of inorganic Sn suffered from anemia, kidney and liver problems. On the other hand, humans exposed to high levels of organotin compounds for short period of time had several health effects which included skin and eye irritation, gastrointestinal effects, respiratory irritation amongst other issues. Ingestion of high amounts of Sn has been reported to lead to certain lethal cases [28]. In this study, the highest and outstanding value for Sn in the nine toothpaste samples was observed in one of the conventional toothpastes (9670.83 ppm) ($p < 0.001$, from the rest). Additionally, the other toothpastes exhibited Sn values above 80 ppm which is much higher than the average values obtained for all the other metals in this study. However, a high Sn value was quoted for cosmetic products in other studies (5795 ppm) [27]. Although Sn is known as a heavy metal, in more recent formulations, it is considered as a desired ingredient. Some toothpastes are typically formulated with Sn in the form of stannous fluoride. A study carried out by Gunsolley [29] indicated that this compound has shown a statistically significant antiplaque effect. The study therefore suggests that the mechanism of action in this compound is its ability to alter plaque reducing its effects on gingivitis [29].

### 3.1.7. Zinc

Zinc is known to be one of the essential elements for humans. Deficiency issues may arise in low intake of this metal. However, like most metals, excess exposure to such elements produces its own toxicological effects. Zinc intoxication is likely to take place via the oral route rather than the inhalation or dermal routes. Excessive ingestion of Zn compounds may result in vomiting, diarrhea abdominal pain and other related gastrointestinal effects. There is also not enough evidence to show that zinc is carcinogenic to humans. However, an elevated intensity of this metal is known to be toxic to living cells in general [30]. In the present study, toothpastes in the conventional category exhibited elevated Zn levels (1842 and 2417 ppm) ($p < 0.001$). These values were significantly higher than those quoted by Odukudu and coworkers [20] in cosmetic products (0.743 ppm). On the other hand, the children's toothpastes did not contain Zn. Zinc is a mineral that is naturally found in plaque, saliva and teeth enamel in the buccal cavity. However, Zn is incorporated in dental cosmetic products to control plaque, prevent the formation of calculus and reduced bad odors. In a study by Lynch [31], it was observed that high Zn levels persist hours after the application of oral hygiene products.

### 3.1.8. Iron

Iron is known to be a vital element for the survival and growth of many living organisms. It forms part of the components making up humans such as oxygen transporting proteins including myoglobin and hemoglobin. The sources of iron contamination in surface water are mainly human activities such as mining. However, excess amounts of this heavy metal in the human body leads to toxic health effects. One of these effects is metabolic acidity, which occurs when elevated amounts of free Fe enters the cells of brain, liver and heart and is converted to ferric iron, releasing hydrogen ions [24]. The content of Fe in toothpastes ranged between 1.76 and 17.68 ppm ($p < 0.001$). The maximum and minimum values were found in the conventional toothpastes. Other studies show that Fe is also present in other cosmetic products. However, the level of Fe is below 1.62 ppm [19,20], which is much lower than those quoted in the present study.

### 3.1.9. Manganese

Manganese is a naturally occurring metal on the earth's crust and is also an essential element of the human diet. Normal dietary intake of Mn is sufficient to maintain suitable levels within the body. However, high levels of Mn can lead to human neurotoxicity. Humans are exposed to Mn via contaminated water, Mn-containing fuel, airborne Mn in work environments, and other

sources. Exposure to excessive amounts can lead to a disease known as manganesism, having similar symptoms to Parkinson's disease [32]. In this study, the highest Mn content was found in the children's toothpastes (2.07 ppm), while the lowest was found in the conventional toothpastes (0.20 ppm) ($p < 0.001$). These values are much lower than those observed by Ideriah and coworkers [19] in toothpastes (12.2 ppm).

### 3.1.10. Cadmium and Mercury

Cadmium was initially used in the paint industry. However, in more recent years, it is being used in rechargeable batteries and the production of alloys, amongst other uses. Cadmium may accumulate in soils for decades until it is taken up by plants, which may be consumed by humans or animals. Cd is known as a nonessential and toxic heavy metal. It has several adverse effects on the human body including the enzymatic systems of cells and also oxidative stress. In the liver, Cd binds with cysteine-rich protein producing a complex which leads to hepatotoxicity that is chemical-driven liver damage. This complex then circulates to the kidney leading to its accumulation and toxicity problems in the renal tissue [24]. Mercury is known to be one of the most toxic metals and is found in the environment from various activities such as agriculture, industrial and wastewater discharge, mining and other sources. Mercury is also used in industry, component of batteries and also as part of the mixture for dental preparations. There are various symptoms which are associated with exposure to metallic Hg which include vomiting, skin rashes, lung damage and others. Inorganic Hg, in the form of $Hg^{2+}$ or $Hg^+$, has a higher solubility in water making its adverse toxicological effects much greater when compared with the elemental mercury. Additionally, due to its lipophilic properties, organic mercury can easily permeate across biological membranes. The major health effects of mercury vary from disruption of the nervous system including adverse effects on the brain to other health issues such as negative reproductive effects which include sperm damage, miscarriages and also birth [26]. In this study, neither Cd nor Hg were detected in the nine toothpaste products. This may be due to possible levels of these metals being lower than the detection limit or due to the absence of these metals. Different studies report different contents for Cd. Rao and Rao [27] did not detect Cd in toothpaste, but other research teams observed levels ranging between 0.467 and 21.42 ppm [18,20,21]. Only one study reports the presence of Hg in toothpaste at a level of 0.06 ppm [27].

### 3.2. Polyphenolic Content

Figure 1 illustrates the total polyphenolic content of the toothpaste samples under study. The highest mean content was found in the herbal toothpaste category (0.135% w/w), while the lowest mean value was obtained in the conventional toothpaste category (0.024% w/w). In spite of these differences, there was no statistically significant differences between the samples ($p > 0.05$). Even though some toothpastes are not herbal in content, the flavor of most toothpastes is minty, giving them the refreshing and pleasant taste. Amongst the flavoring agents used in toothpaste, there are several herbal extracts and oils [33]. These contain phenolic acids, flavones, and flavanones which are claimed to impart antimicrobial and anti-inflammatory activities within the buccal cavity.

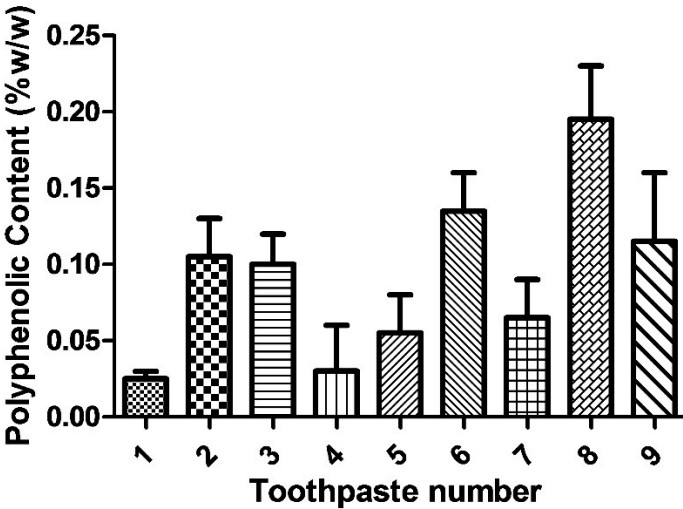

**Figure 1.** The total polyphenolic content in the nine toothpaste samples.

*3.3. Correlation between Presence of Polyphenols and Heavy Metals in Toothpaste*

The presence of polyphenols in toothpastes was correlated with the presence of heavy metals. The correlation analysis was specifically interesting for the category of herbal toothpastes which contain plant extracts as one of their ingredients. The results for the Pearson correlation (Table 5) indicated that the type and quantity of heavy metals is irrespective of the amount of polyphenols found in the nine toothpaste samples. The only metal which gave a weak positive correlation ($r = 0.562$) with the total polyphenolic content was Fe. A similar outcome was also observed in a study, which showed that polyphenols have a chelating ability with Fe, an attribute which is also of importance in the biological systems [34].

**Table 5.** Pearson correlations for heavy metals and polyphenols found in toothpastes.

| Variables | Cr | Cu | Ni | Pb | Sn | Zn | Fe | Mn | PolyP |
|---|---|---|---|---|---|---|---|---|---|
| Ag | 0.037 | 0.115 | −0.201 | 0.526 | −0.044 | 0.340 | −0.513 | −0.867 | −0.337 |
| Cr | | 0.960 | −0.119 | 0.397 | 0.985 | 0.825 | 0.680 | −0.337 | 0.019 |
| Cu | | | −0.097 | 0.521 | 0.958 | 0.799 | 0.548 | −0.310 | −0.073 |
| Ni | | | | −0.617 | −0.160 | 0.032 | −0.113 | 0.312 | 0.127 |
| Pb | | | | | 0.438 | 0.236 | 0.078 | −0.434 | −0.135 |
| Sn | | | | | | 0.769 | 0.723 | −0.226 | 0.096 |
| Zn | | | | | | | 0.236 | −0.538 | −0.221 |
| Fe | | | | | | | | 0.154 | 0.562 |
| Mn | | | | | | | | | 0.262 |

## 4. Conclusions

This study determined the concentration of eleven heavy metals within nine toothpaste samples available at Maltese pharmacies. Pb, Cd, Hg and Cr were compared to international standards for maximum heavy metal content in cosmetics. Most of these heavy metals found in the samples were within the international standards except for certain values of Pb which was not found to be within the US and EU standards. Although authorities worldwide are aware of the potential toxic implications of heavy metal contamination of cosmetics, very few metals are regulated. Some authorities prohibit the presence of these metals but due to the ubiquitous nature of these metals, their presence is unavoidable. One potential problem with toothpastes and other oral hygiene products is that these are borderline products between cosmetics and foodstuffs as they are classified as cleansing agents [35], whereas a quantity of these products may be ingested unintentionally on a daily basis. Therefore, the limits set

for topical cosmetic products may not adequately cover these oral hygiene products. Consequently, in some instances, the limits set for foodstuffs may be more appropriate.

**Author Contributions:** Conceptualization, writing—review, editing and supervision: E.A.; methodology and investigation: A.V.

**Funding:** This research received no external funding.

**Acknowledgments:** The authors acknowledge Adrian Bugeja-Douglas for technical support.

**Conflicts of Interest:** The authors declare no conflict of interest.

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
