# Peer review of "Analysis of Heavy Metal Content in Conventional and Herbal Toothpastes Available at Maltese Pharmacies"

_cosmetics, doi:10.3390/cosmetics6020028_

Round 1

Reviewer 1 Report

This study investigated the presence and heavy metals and polyphenols in toothpaste samples available at Maltese pharmacies. Authors described in a comprehensive manner the significance and the effects of the occurence of heavy metals in cosmetics.On the basis of their results Authors point out that more attention should be paid to the presence of metals in toothpastes, and their content should be monitored constantly. They also suggest that limits of heavy metals tolerated in cosmetics should be equated to those of foodstuffs. In my opinion this paper deserve to be published in  Cosmetics in the present form.

Author Response

Dear Reviewer,

Thank you for your comments.

Sincerely Yours,

Everaldo Attard

Reviewer 2 Report

The manuscript is written at a good scientific level and I recommend it for publication in its current form.

Author Response

(The authors gave the same response as above.)

Reviewer 3 Report

Overall, the manuscript is well written and the topic is of interest to public health. 

Some level of discussion/conclusion should be included in abstract

The significance of a correlation between heavy metal content and polyphenolic content needs to be presented and discussed throughout the manuscript

The authors should discuss the different species of phenols, and whether it's reasonable or not reasonable to study specific species.

Why was ANOVA stat performed for phenols and not metals?

There needs to be a discussion on the impact of the correlation between phenols and metals

Authors should discuss next steps or implications of their research

Author Response

Dear Reviewer,

These are our replies to your comments:

Some level of discussion/conclusion should be included in abstract - we were limited with word count. However, we are including the inadequacy of the current standards to oral hygienic products.

The significance of a correlation between heavy metal content and polyphenolic content needs to be presented and discussed throughout the manuscript- We consider this correlation as a result of the metal analysis and subsequently the polyphenolic content analysis. Therefore, the correlation is the outcome from our work and therefore we thought that this has to be presented at the end of the manuscript.

The authors should discuss the different species of phenols, and whether it's reasonable or not reasonable to study specific species. - As the title implies, we focused mainly on metals. The polyphenols were an added value  to the manuscript and so we did not consider the same depth of analysis for the metals and the polyphenols.

Why was ANOVA stat performed for phenols and not metals?  - Lines 103-105 state that the ANOVA was performed for metals and polyphenols. We have now included the p values for each metal.

There needs to be a discussion on the impact of the correlation between phenols and metals - section 3.3 lines 278-285 are dedicated to this correlation. As we mentioned before, the polyphenolic content of toothpastes was a secondary aim, which ultimately would merit a whole research manuscript on this topic.

Authors should discuss next steps or implications of their research - Section 4 lines 289-301, conclude on the importance of such findings within the set up of international standards. However, this study criticizes the way such oral hygienic products are inappropriately fitted within standards which pertain either to topical (dermal) cosmetic products or else systemic (oral) food products.

We hope that we have satisfactorily amended the manuscript and replied to your queries, according to your comments.

Thank you,

Sincerely Yours,

Everaldo Attard